# Ambient Air Pollution, Extreme Temperatures and Birth Outcomes: A Protocol for an Umbrella Review, Systematic Review and Meta-Analysis

**DOI:** 10.3390/ijerph17228658

**Published:** 2020-11-21

**Authors:** Sylvester Dodzi Nyadanu, Gizachew Assefa Tessema, Ben Mullins, Bernard Kumi-Boateng, Michelle Lee Bell, Gavin Pereira

**Affiliations:** 1Education, Culture and Health Opportunities (ECHO) Ghana, ECHO Research Group International, P. O. Box 424, Aflao, Ghana; 2School of Public Health, Curtin University, Perth, Kent Street, Bentley, Western Australia 6102, Australia; gizachew.tessema@curtin.edu.au (G.A.T.); b.mullins@curtin.edu.au (B.M.); gavin.f.pereira@curtin.edu.au (G.P.); 3School of Public Health, University of Adelaide, Adelaide, South Australia 5000, Australia; 4Department of Geomatic Engineering, University of Mines and Technology, P.O. Box 237, Tarkwa, Ghana; kumi@umat.edu.gh; 5School of the Environment, Yale University, New Haven, CT 06511, USA; michelle.bell@yale.edu; 6Telethon Kids Institute, Northern Entrance, Perth Children’s Hospital, Nedlands, Western Australia 6009, Australia; 7Centre for Fertility and Health (CeFH), Norwegian Institute of Public Health, 0473 Oslo, Norway

**Keywords:** ambient air pollution, temperature, birth outcomes, perinatal outcomes, umbrella review, systematic review, meta-analysis, low and middle-income countries, LMICs

## Abstract

Prenatal exposure to ambient air pollution and extreme temperatures are among the major risk factors of adverse birth outcomes and with potential long-term effects during the life course. Although low- and middle-income countries (LMICs) are most vulnerable, there is limited synthesis of evidence in such settings. This document describes a protocol for both an umbrella review (Systematic Review 1) and a focused systematic review and meta-analysis of studies from LMICs (Systematic Review 2). We will search from start date of each database to present, six major academic databases (PubMed, CINAHL, Scopus, MEDLINE/Ovid, EMBASE/Ovid and Web of Science Core Collection), systematic reviews repositories and references of eligible studies. Additional searches in grey literature will also be conducted. Eligibility criteria include studies of pregnant women exposed to ambient air pollutants and/or extreme temperatures during pregnancy with and without adverse birth outcomes. The umbrella review (Systematic Review 1) will include only previous systematic reviews while Systematic Review 2 will include quantitative observational studies in LMICs. Searches will be restricted to English language using comprehensive search terms to consecutively screen the titles, abstracts and full-texts to select eligible studies. Two independent authors will conduct the study screening and selection, risk of bias assessment and data extraction using JBI SUMARI web-based software. Narrative and semi-quantitative syntheses will be employed for the Systematic Review 1. For Systematic Review 2, we will perform meta-analysis with two alternative meta-analytical methods (quality effect and inverse variance heterogeneity) as well as the classic random effect model. If meta-analysis is infeasible, narrative synthesis will be presented. Confidence in cumulative evidence and the strength of the evidence will be assessed. This protocol is registered with PROSPERO (CRD42020200387).

## 1. Introduction

Air pollution and extreme temperatures (heat/cold waves) are ubiquitous exposures that may explain a fraction of adverse birth outcomes (e.g., preterm birth, stillbirth and foetal growth restriction), pregnancy complications (e.g., miscarriage, pre-eclampsia and prelabour rupture of membranes) and longer-term effects (e.g., neurological, hormonal, respiratory and cardiovascular disorders) [1,2,3,4]. Environmental hazards contribute substantially to public health emergencies [5], with one in every nine deaths attributable to air pollution, ranking as the fifth leading risk factor of mortality [5,6]. Some common health-damaging air pollutants are gaseous air pollutants such as nitrogen dioxide (NO_2_), carbon monoxide (CO), ozone (O_3_), sulphur dioxide (SO_2_), polycyclic aromatic hydrocarbons (PAH) [1,7,8] and particulate matter (PM), including those with aerodynamic diameter ≤2.5 μm (PM_2.5_) and ≤10 μm (PM_10_) [9]. Although biological mechanisms are not fully established, there is accumulating evidence indicating that environmental hazards (e.g., air pollutants and extreme temperatures) might alter and trigger a cascade of pathophysiological responses, especially excess oxidative stress, and cardiovascular, immuno-inflammatory and metabolic alterations which affect prenatal development [10,11]. These patho-aetiological processes result in adverse reproductive outcomes which are exacerbated by obstetric or maternal health conditions, biologic, sociodemographic and behavioural risk factors [12,13,14].

With the increasing volume of relevant literature and the need for an understanding of the overall scientific evidence, systematic reviews and meta-analyses objectively synthesise scientific evidence to address environmental health questions [15], for informed decision-making by health practitioners, policy makers and other stakeholders [16,17,18]. Despite the mixed findings, syntheses of available literature have indicated possible associations between ambient air pollution and birth outcomes [19,20,21,22,23,24,25]. The literature that has examined extreme temperatures (heat/cold waves) and birth outcomes in original studies [26,27,28] and reviews [19,29,30,31] have also supported the hypothesis of positive associations. Systematic reviews and meta-analyses are crucial in harmonising the evidence but similar to original primary studies, they also have varied scope, quality and conclusions [32], and therefore the challenge of making evidence-based informed decisions resurfaces as reviews accumulate [16,33]. It is therefore prudent, logical and recommended [16] to perform umbrella reviews, a systematic synthesis of evidence from existing systematic reviews and meta-analyses [16,17]. A recent overview of meta-analyses on occupational exposures and pregnancy outcomes was conducted, concluding that maternal exposures to harmful substances can lead to many adverse pregnancy outcomes and birth defects [34]. Similar reviews of reviews (i.e., umbrella reviews) have been conducted for other exposures associated with birth outcomes, such as antenatal depression [35] and periodontal disease [36], but we are not aware of an equivalent study for associations between ambient environmental exposures, such as air pollution and/or extreme temperatures and birth outcomes. We conducted a preliminary search of PubMed and PROSPERO, which revealed one study that synthesised meta-analyses on environmental risk factors and pregnancy outcomes [32]. That study included only one meta-analytical result [25] on ambient air pollution and adverse birth outcomes and also noted that most meta-analyses did not follow meta-analysis methodological guidelines [32]. For ambient air pollution and birth outcomes, numerous systematic reviews and meta-analyses [19,20,21,22,23,24,25] have reported consistent positive associations and conclusions but with both statistically non-significant [20,23] and significant [22,25] associations of PM_2.5_ with preterm birth (PTB), statistically non-significant [21,25] and significant [20] associations of PM_10_ with low birthweight (LBW) and statistically significant [22] and non-significant [20] associations of O_3_ with PTB. Although the conclusions are consistent across the recent reviews on the increased risk of exposure–cause–effect, the mixed statistical significance and the varied scope of the reviews are likely to be perceived by policy makers or other stakeholders as confusing, resulting in delay in timely intervention. Similarly, variations in temperature metrics hindered meta-analysis in this domain but few systematic reviews without meta-analysis [19,30,31] from this relatively new and emerging area of research have also indicated negative impacts of extreme ambient air temperatures on pregnancy outcomes. Evaluating the importance and strength of the evidence through a well-planned umbrella review is now required to systematically and comprehensively synthesise the numerous existing systematic reviews and/or meta-analyses to inform current policies, to provide an explanation for associations and to inform future research directions [16,33].

The evidence in the existing reviews on ambient air pollution and/or temperature and birth outcomes [19,20,21,22,23,24,25,30,31,37] is heavily based on studies for high income countries while acknowledging lack of evidence from low-and middle-income countries (LMICs). Conceivably, this may be due to generally limited environmental health researches in developing countries such as Africa [38,39] and even in some East Asian and Pacific Island countries. Despite their limitations, analytical cross-sectional and ecological studies provide exploratory information to generate hypotheses for possible links between environmental factors and disease outcomes [18,40]. However, these study designs were excluded in previous reviews [20,21,24] which could lead to excluding evidence from under-resourced settings. The Preferred Reporting Items for Systematic reviews and Meta-Analyses (PRISMA) [41] and the proposal for Meta-analysis Of Observational Studies in Epidemiology (MOOSE) [42] recommend broadening inclusion criteria to include most studies while implementing sensitivity and/or stratification. Stringent inclusion/exclusion criteria will improve the homogeneity among primary studies for valid cause-and-effect reviews, but this can limit the external generalisability and applicability of the findings [43]. Acknowledging the potential of these study designs in shedding light on the exposure-outcome association, recent reviews in environmental and occupational health are increasingly including ecological and analytical cross-sectional studies [22,23,44,45,46]. Moreover, searching databases alone is not necessarily sufficient to retrieve relevant studies [47] from LMICs and some reviews searched one [22] or two [20,25] databases. Notably, grey literature sources were not searched in previous reviews, which could also lead to missing yet relevant studies [43,47] from LMICs. A recent exploratory study on optimal database combinations for literature searches in systematic reviews concluded that optimal literature searches must search MEDLINE, EMBASE, Web of Science and Google Scholar (the first 200 relevant references) as a minimum requirement and any special topic databases to optimise adequate and efficient coverage of locating relevant studies [47].

Many would accept that populations within LMICs are possibly the most vulnerable to the effects of such exposures on perinatal endpoints given their already elevated health burden [48,49] but the quality of air in most LMICs is not monitored reliably as compared to high-income countries (HICs). Consequently, one conclusion made by Rees et al. [49] is that “we are not only potentially underestimating the impact we might also not know how bad it is until it is too late”. Results from recent studies [38,39,50] using Demographic Health Survey (DHS) data and gridded satellite-based estimates of PM_2.5_ across Africa indicate strong significant associations. For instance, exposure to early-life carbonaceous PM_2.5_ increased the odds of neonatal mortality (OR: 1.22; 95% CI: 1.11–1.35) on the log PM_2.5_ exposure level [38], higher odds for pregnancy loss cases with 26.64 μg/m^3^ exposure than the control with 25.69 μg/m^3^ exposure level (1.22; 1.107–1.137), and this included miscarriage (1.125; 1.109–1.142) and stillbirth (1.094; 1.05–1.38) per 10 μg/m^3^ increase in PM_2.5_ exposure [39]. Some researchers [38] have suggested lowering World Health Organization (WHO) air quality guidelines below the current 10 μg/m^3^ total mass guideline for harmful carbonaceous PM_2.5_ excluding dust and sea-salt levels [38]. Although studies from China are comparatively well-represented in previous reviews [20,51,52], relatively recent studies have been conducted in other LMICs such as India [53], South Africa [54] and across 33 African countries (using 68 surveys from 1998–2016) [39]. The tropical climatic zone of Sub-Saharan Africa adds to the impacts of extreme temperatures in these settings. A focussed systematic review and meta-analysis in LMICs on ambient air quality and temperature and the risk of adverse birth outcomes is required for these most vulnerable settings. A similar review, focussing on LMICs was planned for household air pollution and birth outcomes elsewhere [55].

The Grading of Recommendations Assessment, Development and Evaluation (GRADE) system is widely used in systematic reviews and meta-analyses, health technology assessment and clinical practice guidelines [56] and adopted by several national and international organisations [57,58]. However, direct utility of GRADE in environmental and occupational health reviews is challenging [58,59], which could have contributed to inability to evaluate the confidence in cumulative evidence in the previous reviews [19,20,21,22,23,24,25,30,31,37]. Fortunately, the Navigation Guide systematic review methodology refined GRADE for environmental health risk assessment of human observational studies [60] as reported recently [61,62]. A recent WHO review on effects of environmental noise on cardiovascular and metabolic diseases also modified GRADE [59] and such modifications have been applied elsewhere [44,45,46,63]. Thus, there is an opportunity to rate the confidence in cumulative evidence on the effects of air pollution and/or temperature by adapting the modified GRADE system [59,60] as well as translating the overall confidence into plausible toxicological effects per Navigation Guide criteria [61,62].

The aims of this study are therefore: (i) to systematically and comprehensively examine and synthesise the literature on the effects of ambient air pollution (and if reviews are available, temperature) on birth outcomes via umbrella review (Systematic Review 1); and (ii) to use first-order systematic review and meta-analysis to systematically synthesise the available evidence on the topic in the most vulnerable settings, LMICs (Systematic Review 2). Overall, this will improve knowledge of the associations between ambient air pollution and temperature and birth outcomes globally, provide an evidence-base to inform decision making and identify gaps for further research.

## 2. Materials and Methods

This systematic review protocol was developed using the statement and checklist of Preferred Reporting Items for Systematic reviews and Meta-Analyses for Protocols (PRISMA-P) [64,65]. The conduct of the systematic review and meta-analysis will be guided by the PRISMA statement [66], the proposal for Meta-analysis Of Observational Studies in Epidemiology (MOOSE) [42] and Joanna Briggs Institute (JBI) systematic reviews collaboration [33]. This review will include a comprehensive synthesis of evidence from existing systematic reviews and meta-analyses through an umbrella review approach (Systematic Review 1) and systematically evaluate the primary evidence from LMICs (Systematic Review 2)

### 2.1. Eligibility Criteria

Eligible studies in this review will address the objectives of the review according to the PECOS (Participants, Exposures, Comparators, Outcomes and Study design) statement [60,61] recommended for environmental and occupational health research.

#### 2.1.1. Participants or Populations

The participants are pregnant women and foetuses (*in-utero* infants) at any period of pregnancy up to birth.

#### 2.1.2. Exposures

The exposures to be included in this study are prenatal exposure to ambient (outdoor) air pollution and/or ambient air temperature. The most commonly used markers of ambient air pollution, nitrogen dioxide (NO_2_) or nitrogen oxides (NOx), carbon monoxide (CO), ozone (O_3_), sulphur dioxide (SO_2_) [1,7,8], fine particulate matter (PM) at aerodynamic diameter ≤2.5 μm (PM_2.5_) and coarse particles ≤10 μm (PM_10_) or total suspended particles (TSP) [9] will be considered and as non-occupational exposures. Studies on temperature and birth outcomes used different metrics such as threshold temperature (mean or percentile with different durations), maximum temperature, heat-humid index, thermal heat sensation and heat index [30,31,67]. All reported metrics for temperature will be considered.

#### 2.1.3. Comparators

The comparators (control groups) are pregnant women in the same study population and period with lower exposure levels with or without adverse birth outcomes as compared to those exposed to higher exposures with adverse birth outcomes.

#### 2.1.4. Outcomes

The adverse perinatal outcomes of interest include: preterm birth (PTB; live birth before 37 completed gestational weeks, pregnancy loss (miscarriage and stillbirth), birth weight and foetal growth restrictions (term low birth weight, TLBW or LBW: birth weight <2500 g at ≥37 completed gestational weeks; and small-for-gestational age; SGA: birth weight below the 10th percentile for that gestational age and sex; and foetal or intrauterine growth restriction).

#### 2.1.5. Study Designs

For both systematic reviews, we will include only quantitative human observational studies: prospective/retrospective cohort, case-control, analytical cross-sectional and ecological studies that examined long-term effects (that is, entire pregnancy or by trimesters) of ambient air pollution and/or temperature on birth outcomes. The analytical studies assessing short-term effects (e.g., last month of gestation and few weeks or days to birth), including daily time series and case-crossover studies, will be included and synthesis will be performed separately by exposure period. Randomised controlled trials (RCTs) are impractical in this domain, but any RCTs and/or natural human experiments will be included if identified.

Systematic Review 1(umbrella review) will include all systematic reviews with or without meta-analyses irrespective of geographical location or economic grouping. A systematic review and/or meta-analysis will be included if the review study specified inclusion/exclusion criteria, specified a search strategy in at least one literature database, clearly reported results on any of the exposure–outcome associations of interest (as defined in our PECO statement) as primary objective with details on the included primary studies [68] and included at least three primary studies for the exposure-outcome association of interest [69].

#### 2.1.6. Exclusion Criteria

For both systematic reviews, studies investigating other reproductive health outcomes (e.g., pre-eclampsia) and studies using only distance from/to the source of exposure (e.g., distance to road) as proxy without empirical assessment of exposures will be excluded. Descriptive epidemiological studies (e.g., case reports, case series and descriptive cross-sectional), studies without full data/report (e.g., conference abstracts, letters to the editor and editorials), non-human studies (e.g., animal model and *in vitro*) and assisted-reproductive technology (e.g., *in vitro* fertilisation and embryo transfer) will be excluded.

Systematic Review 1(umbrella review) will exclude theoretical reviews or reviews incorporating theoretical studies or opinion as primary source of evidence.

Systematic Review 2 (systematic review and meta-analysis in LMICs) will include only primary studies conducted on participants from LMICs, using the current World Bank economic grouping [70]. No exclusion criterion will be applied to adjustment of confounding factors, but we will summarise the confounders adjusted in each study. Multi-country study that included both LMICs and high-income countries (HICs) will be selected but data will be retrieved for the included LMICs only.

### 2.2. Information Sources

Both published and grey literature will be sourced from: (i) six major bibliographic databases (PubMed, CINAHL, Scopus, MEDLINE via Ovid, EMBASE via Ovid and Web of Science Core Collection); (ii) systematic reviews repositories (Cochrane Database of Systematic Reviews, JBI Database of Systematic Reviews and Implementation Reports, and Epistemonikos; www.epistemonikos.org/); (iii) electronic grey literature databases, OpenGrey (http://www.opengrey.eu/) and WorldWideScience.org; (iv) Internet search engines, Google (www.google.com/) and Google Scholar (www.google.com/scholar/), to screen the first 200 hits for potentially relevant studies [47]; (v) World Health Organisation website; and (vi) references of eligible studies. Searches will be restricted to English language with no date limitations. The dates of searches will be recorded.

### 2.3. Search Strategy

We searched medical subject headings (MeSH) with key words related to the exposures (ambient air pollution and temperature) and the adverse birth outcomes based on terminologies used in recent reviews on the topic [19,20,21,22,23,24,25,29,30,31,37]. Comprehensive search terms using the relevant MeSH terms, key words and previous search terms will be developed (e.g., air pollution, particulate matter, temperature, climate change, heat, pregnancy outcome, birth outcome, birth weight, foetal growth, stillbirth, premature birth, preterm birth and small-for-gestational age). These search terms will be used within PubMed as template database to finalise an advanced search strategy using Boolean combination and will be modified where necessary for the rest of the databases and the other sources. The search terms within each search grid category will be expanded with “OR” and the two categories combined with “AND” to search in the “Title/Abstract”. Example for PubMed is given in Appendix A. For the umbrella review (Systematic Review 1), additional search terms “review” and “meta-analysis” will be applied to obtain previous systematic reviews and meta-analyses. A librarian from Faculty of Health Sciences, Curtin University with expertise in searching databases for systematic reviews will be consulted on the search strategies for each database. Reference lists of the eligible primary studies and previous reviews will also be searched manually to further identify potentially eligible studies that might be missed from the database literature search. Alerts will be set for each database, and we will also conduct literature search in PubMed and Scopus for the most recent publications as e-print or in-press ahead of publication over the last four months when we are close to completing the review. The dates of searches, including the last search, will be recorded.

### 2.4. Study Screening and Selection

All stages will be conducted independently by two researchers, with conflicts managed by discussion or with a third author. From the search, the titles of all identified citations with abstracts will be uploaded into *EndNote* library and duplicates removed. We will screen the title and abstract per the eligibility criteria. Potentially eligible studies will be retrieved and imported into Joanna Briggs Institute System for the Unified Management, Assessment and Review of Information (JBI SUMARI) web-based software [33,71]. The web-based all-in-one new JBI SUMARI software for conducting all types of review will be used to facilitate the review process [71]. The full text of the selected studies will be assessed comprehensively against the inclusion criteria within the JBI SUMARI system. All studies that do not meet the inclusion criteria will be excluded with reasons and presented in PRISMA flow chart [41]. Erratum or retraction status of the selected studies will be checked.

### 2.5. Quality (Risk of Bias) Assessment of Selected Studies

The methodological quality or risk of bias (RoB) of all selected eligible studies (previous systematic reviews and meta-analyses as well as the primary studies in LMICs) will be assessed by two authors independently with conflicts resolved by consensus or with a third investigator. A study design-specific standardised critical appraisal tools in JBI SUMARI software [71], detailed in the JBI reviewer’s manual [33] will be used. The critical appraisal checklists have series of items to be checked as “yes”, “no”, “unclear” and rarely “not applicable”. To rate the overall RoB of each study in this review, we will assign a score of 1 if a criterion is met (yes) and 0 if the criterion is either not met (no) or lack enough information to judge (unclear). The scores will be summed where high score indicates high quality or low RoB and vice versa.

#### 2.5.1. Systematic Review 1 (Umbrella Review)

For the umbrella review (11 items), scores 0–5 will be classified as low quality (high RoB), 6–8 as moderate quality (moderate RoB) and 9–11 as high quality (low RoB) for the previous systematic reviews and/or meta-analyses. We will further assess the methodological quality of the included reviews with the revised AMSTAR (A MeaSurement Tool to Assess systematic Reviews, AMSTAR 2) critical appraisal tool [72] to clearly identify critical flaws in specific critical domains in rating the overall confidence in the results of each systematic review and/or meta-analysis as “high”, “moderate”, “low” and “critically low” (Appendix A) [72].

#### 2.5.2. Systematic Review 2 (Systematic Review and Meta-Analysis in LMICs)

Using the JBI critical appraisal checklists within the JBI SUMARI web-based software [71], cohort studies (11 items) of scores 0–5 will be classified as low quality (high RoB), 6–8 as moderate quality (moderate RoB) and 9–11 as high quality (low RoB); case-control studies (10 items) classified with 0–4, 5–7 and 8–10 scores for low, moderate and high quality, respectively; and cross-sectional (8 items) with 0–3, 4–6 and 7–8 scores for low, moderate and high quality, respectively. To our knowledge, critical appraisal checklists for ecological studies are not available and will be considered low quality by default (high RoB). If required, corresponding author(s) will be contacted for additional information for clarification. In such case, at least two attempts will be made to contact the corresponding author.

There are emerging and substantially consistent contextualised RoB criteria for observational human studies in environmental and occupational health [73]. We therefore modified an updated WHO evidence review’s RoB for noise pollution and birth outcomes [44] by using information from Navigation Guide [61,62] and MOOSE [42] to obtain a precise and concise but comprehensive, transparent, reproducible, and objective RoB criteria (Appendix A) for additional appraisal of the RoB of the primary studies in similar fashion [44,45]. The score for each domain will be presented and the overall RoB scores to rate each study as high quality (low RoB) if total score is 26–33 (at least 80% of maximum score 33), moderate quality (moderate RoB) if 17–25 (less than 26 but ≥50% of 33) and low quality (high RoB) for <17 (<50% of 33). All eligible studies will be included in the data synthesis irrespective of the results of the RoB [73] due to the non-consensus around quality rating and as recommended by the MOOSE group [42]. However, because results from subgroup and sensitivity analyses by RoB may lead to inconsistent results or spurious associations within strata due to collider-stratification bias [74], the proposed improved alternative meta-analytic quality effect model will be performed to account for RoB variance [74,75].

### 2.6. Data Extraction and Management

#### 2.6.1. Systematic Review 1 (Umbrella Review)

Two authors will extract data from the selected studies with a data extraction tool (Appendix A) developed according to the relevant data for the umbrella review as summarised in Table 1. The data extraction tool will be piloted prior to the full data extraction process. Any disagreements between the two authors will be resolved by consensus or with a third author.

#### 2.6.2. Systematic Review 2 (Systematic Review and Meta-Analysis in LMICs)

Similarly, data will be extracted by two authors with a piloted data extraction tool (Appendix A) according to the key data elements for the Systematic Review 2, as summarised in Table 2.

Results will be extracted for estimated effects for each criteria air pollutant and temperature separately for each study and LMIC-specific results for multi-country study that included both LMICs and HICs. If required, author(s) will be contacted for missing or additional data or to clarify the existing data through an email with two follow-up emails. In some cases, multiple studies may use the same underlying data (e.g., follow-up study). In this case, the most extensive data on the main findings of the study will be selected.

### 2.7. Data Synthesis and Statistical Analysis

#### 2.7.1. Systematic Review 1 (Umbrella Review)

General characteristics and scope of each review will be summarised based on the data extracted (Table 2) using tables and figures with textual descriptions. Structured tabular and pictorial groupings of reviews will also be presented based on the meta-analytical model used (fixed and/or random); heterogeneity test (Cochran’s Q and/or I^2^); heterogeneity level (low, moderate, high or larger I^2^ > 50); study period (categories of five-year intervals); number of databases used (1, 2–3 or >3), number of studies included in meta-analysis (≤5, 6–10, 11–20 or >20); and yes/no for searched grey literature, registered/published protocol, assessed and rated quality of included studies with a RoB tool, followed systematic review and/or meta-analysis guidelines, checked publication bias and the tests used and performed subgroup/sensitivity analyses. The methodological quality of each review will also be presented.

Following JBI umbrella review methodology, the core rationale for conducting an umbrella review is to systematically summarise the evidence from multiple top-tier bodies of evidence (systematic reviews and/or meta-analyses) on a given topic but not to re-synthesise the results of the previous reviews or synthesis with meta-analysis or meta-synthesis [16,33]. Thus, without statistically pooling the results of previous systematic reviews and meta-analyses, we will adapt a semi-quantitative approach to systematically evaluate the confidence in cumulative evidence across previous systematic reviews with meta-analyses as reported in other umbrella reviews [76,77,78]. The updated two grading scales in [76] will be used to judge the importance of each exposure at six levels (Appendix A) and the strength of the evidence in terms of consistency in the findings of previous systematic reviews with meta-analyses and quality of included study designs at four levels as “convincing evidence” (CE), “probable evidence” (PE), “limited-suggestive evidence” (LSE) and “limited, no conclusive evidence” (LNCE) (Appendix A)**.** Combining the two grading scales will give overall epidemiological evidence of plausibility or not for a cause-and-effect association.

#### 2.7.2. Systematic Review 2 (Systematic Review and Meta-Analysis in LMICs)

A minimum of five comparable studies for a birth outcome with adequate quantitative data will be required to conduct a meta-analysis for that birth outcome, otherwise only deep narrative synthesis will be undertaken. To be comparable, studies must address the same exposure timeframe (e.g., whole pregnancy), birth outcomes (e.g., preterm birth) and exposure (e.g., PM_2.5_). The narrative synthesis will include summarising the characteristics of the study population, methodological quality, exposure measurements, birth outcome assessments, confounders adjusted and statistical significance of the effect estimates.

##### Main Meta-Analysis

We will conduct the meta-analysis and examine the potential publication bias with an open access meta-analysis package MetaXL version 5.3 [79]. The two novel meta-analytical models, the inverse variance heterogeneity (IVhet) and quality effect (QE) models, which use quasi-likelihood-based variance structures with no distributional assumptions [74,75,80,81,82,83], will be used. Unlike the random effects (RE) model, both IVhet (a modified fixed effect model) and the QE model favour large studies regardless of increasing heterogeneity and the robust QE model (a bias adjustment method without bias quantification but computes synthetic bias from the quality score) additionally favours studies with better methodological quality [75,80,81]. Comparatively, IVhet and QE models were demonstrated to outperform the conventional (random effect and fixed-effect) models with higher precision and probability of producing estimates closer to true effect sizes [74,75,80,81,82,83,84] and QE estimators also bypass collider-stratification bias (induced by stratification or meta-regression or leave-one-out sensitivity analyses based on RoB results) [74,75]. Several recent meta-analyses [44,45,85,86] have applied IVhet and QE models. The QE model will be used to report the main findings while supplementing results based on IVhet and RE models. We will pool the effect estimates for the entire pregnancy exposure period, and, if data allow, pooled estimates for specific exposure periods (trimester-specific and short-term measures) will also be performed. Because the effect estimates of dichotomous outcomes are mostly expressed as odd ratios (ORs), any relative risk (RR) reported will be converted to OR with the algorithm described elsewhere [87]. Overall (average) exposure levels for each criteria air pollutant and temperature for each study (and, if available, LMIC-specific for multi-country studies that included LMICs) will be summarised. For comparability across studies, a common reference scale of effect estimates will be calculated for increase in exposure per 10 μg/m^3^ for PM_2.5_ and PM_10_; 10 part per billion (ppb) for nitrogen dioxide (NO_2_), NOx and ozone (O_3_); 5 ppb for sulphur dioxide (SO_2_); and 1 part per million (ppm) for carbon monoxide (CO) as described previously [22]. The pooled effect sizes will be expressed as odd ratios (ORs) or hazard ratios (HRs) for dichotomous outcomes and weighted or standardised mean differences or linear regression beta coefficients for continuous outcomes. Forest plots will be used to visually summarise effect estimates with 95% confidence intervals (CIs). Statistical heterogeneity across studies will be evaluated with Cochran’s *Q* statistic at *p* < 0.1 and percentage of inconsistency quantified with *I*^2^ statistic, where 25%, 50% and 75%, respectively, indicate low, moderate and high degree of heterogeneity [88]. The statistical significance for the pooled effect estimates will be two-sided at *p* < 0.05.

##### Subgroup Meta-Analyses and Meta-Regression

Where data permit, series of subgroup analyses will be performed (and if possible meta-regression). This will include: study period (categories of five-year intervals), study region (e.g., Africa, Asia and Caribbean), country with the largest number of studies versus others, study designs (or longitudinal versus non-longitudinal), sample size (four categories), mean exposure levels (four categories), exposure data source and exposure assessment methods (e.g., monitor, modelled, satellite imagery and hybrid method) [37] and level of confounders adjusted for. We will also perform subgroup analysis by World Bank’s economic group (low, lower-middle and upper-middle) [70], global gender gap index on the scale of 0 (inequality) to 1 (full gender equality) (0.0–0.2, worst; 0.3–0.5, bad; 0.6–0.8, good; and 0.9–1.00, best) [89], country’s hunger and nutritional status with the global hunger index by severity scale (≤9.9, low; 10.0–19.9, moderate; 20.0–34.9, serious; 35.0–49.9, alarming; and ≥50.0, extremely alarming) [90], political stability index [91] (≤ −2.0, extremely weak; −2.0< to ≤ −1.0, very weak; −1.0 < to ≤ 0.0, weak; 0.0 < to ≤ 1.0, moderate; 1.0 < to ≤ 2.0, strong; and >2, very strong) and climatic zone (tropical, subtropical, temperate and polar/cold) [92].

##### Sensitivity Meta-Analyses

We will also evaluate the robustness of the result through leave-N-out sensitivity analyses by repeating analyses after removing N studies (e.g., outlying studies, studies with largest or smallest effect estimates and sample sizes and region with largest number of studies). This will indicate which single or combination of studies is/are primarily responsible for between-study heterogeneity.

##### Publication Bias

Potential publication and other forms of bias will be checked with the Doi plot and Luis–Furuya–Kanamori (LFK) index to detect and quantify asymmetry of the study effects in the Doi plots [93]. The Doi plots and LFK index were demonstrated to have greater power/sensitivity than the classic funnel plots and Egger’s test, particularly obvious when the number of studies is small [79,84,93], and have been used elsewhere [44,45,85,86]. We will also report the funnel plots and Egger’s regression *p*-values.

##### Confidence in Cumulative Evidence across Studies

Following the WHO contextualised version of GRADE for environmental and occupation health reviews [46,59] as applied in related reviews [44,45,63,94], we will determine the initial level of quality of evidence across studies based on the study designs and subsequently downgrade by considering GRADE criteria [56,95]: (i) the risk of bias across studies; (ii) inconsistency of results; (iii) indirectness of evidence; (iv) imprecision of the effect estimate; and (v) publication bias or evidence from only one high quality study. We will upgrade for: (i) large magnitude of effect estimate (RR > 1.5) [59]; (ii) a study reporting an association in the presence of accounting for all plausible residual confounders; and (iii) evidence of exposure dose–response gradient. The confidence in the cumulative evidence for each exposure and each birth outcome will be rated as “high”, “moderate”, “low” and “very low” certainty. We will apply Navigation Guide systematic review criteria [60,61,62] to translate the confidence in cumulative evidence ratings into strength of evidence of the health effect as “sufficient evidence of toxicity”, “limited evidence of toxicity”, “inadequate evidence of toxicity” and “evidence of lack of toxicity” (Appendix A). Two authors will carry out the rating and discrepancies resolve by discussion or with a third author.

### 2.8. Ethics and Dissemination

Ethical approval is not required for this review of previously published studies. The findings will be disseminated by publication in peer-reviewed journals and/or conference presentations.

### 2.9. Review Registration

This review protocol was registered with International Prospective Register for Systematic Reviews (PROSPERO) under the identification code CRD42020200387.

### 2.10. Protocol Update

Any necessary amendments in the methods of the present protocol will be updated in PROSPERO and subsequently documented in the final reports with appropriate justifications for the amendments under the caption “Protocol Amendments”.

### 2.11. Limitations of This Study

The meta-analysis for all exposure types and some planned subgroup or sensitivity analyses or meta-regression might not be performed due to potential small number of studies in LMICs. Due to diversity in extreme temperature metrics in the literature, results might not be combinable statistically. The English language restriction could result in missing some relevant studies, but this systematic bias is extremely minimal [96]. As a known limitation of umbrella reviews, a primary study has the potential to be reported in more than one review. However, this will be summarised and reported by computing the Corrected Coverage Area (CCA) index proposed by Pierper et al. [68] and considered in the interpretations.

### 2.12. Strengths of This Study

This is the first umbrella review planned to systematically synthesise and evaluate the epidemiological strength of evidence from existing systematic reviews and meta-analyses on the topic. This review will also specifically and rigorously examine evidence from the most vulnerable regions, LMICs. The use of new improved meta-analytic models (quality effects and inverse variance heterogeneity models) and identification of publication bias will improve the inference if the number of the included studies is small [74,75,80,81,82,83,84,93]. In addition to the general risk of bias (RoB) scales, this study will also assess the RoB in the primary studies using environmental exposure-outcome oriented RoB tool. Furthermore, unlike the previous reviews, this review will evaluate the confidence of the body of evidence with the WHO evidence review’s modified GRADE [59] and also grade the plausible toxicological strength of the evidence according to Navigation Guide principle [60,61,62].

## 3. Conclusions

The prenatal stage is a very sensitive period for development in the life course and exposure to occupational and environmental hazards can have immediate and long-term negative impacts on the offspring [34]. The scientific evidence on environmental hazards (such as ambient air pollution and temperature) and reproductive health outcomes is large, of variable quality and largely unfamiliar to policy-makers, healthcare givers and patients; and compounded with no clear-cut roadmap for evidence evaluation, which could be impediments to timely evidence-based advice on preventive measures, including regulatory measures [57]. Many systematic reviews and/or meta-analyses on the topic with varied scope and quality have accumulated with similar conclusion on the adverse effect of the environmental exposures on perinatal outcomes but with differing statistical significance. Moreover, almost all the primary studies included in the previous reviews were from high-income countries and there is no clear information from the most vulnerable settings, LMICs. In addition to studies from China often captured in the previous reviews, recent studies [39,53,54] are emerging from other LMICs. Hence, synthesis of evidence in LMICs is now possible.

Employing an umbrella review to comprehensively synthesise existing systematic reviews and/or meta-analysis and then purposely pooling the evidence in LMICs with novel approaches, including improved robust meta-analytical QE model [74,75,81], grading the overall evidence with modified GRADE [59] as in previous environmental meta-epidemiology [44,45,46,63,94] and rating the strength of the cause-and-effect per Navigation Guide criteria [60,61,62] will contribute significantly to improved knowledge and inform future studies. We expect that this protocol will provide a succinct outline for searching, extracting and synthesising the relevant information.

## Figures and Tables

**Table 1 ijerph-17-08658-t001:** Key data extraction elements for Systematic Review 1 (umbrella review).

Data Element	Key Indicators
Publication data	First author, journal, publication date, number of citations (to be determined from Google Scholar prior to final data synthesis)
Aims and type of review	Aims/objectives, review type (systematic review, meta-analysis and systematic review and meta-analysis)
Literature search	Number and names of databases searched, date range of databases searched, language restriction and non-databases searched. Review guideline(s) used
Included primary studies	Number of included primary studies, country/continent of the included studies, number of each type of study design included and publication date range of included primary studies
Participants, exposures and outcomes	Total participants included in the review, description of study participants, exposure and outcome assessments
Risk of bias assessment	Risk of bias tool used to appraise the primary studies and the quality ratings
Data synthesis and results	Method of data synthesis, overall results (for meta-analyses, this will include pooled effect sizes and confidence intervals for whole pregnancy and/or trimester-specific or short-term period for air pollutants and any timeframe reported for extreme temperatures, heterogeneity measures, *p*-values and publication bias test, any reported estimates for subgroup/sensitivity analyses)
Conclusion, recommendations and limitations	Researchers’ summary statement/conclusion or interpretation of the main findings (particularly from the abstract), overall recommendations and limitations
Funding and Conflict of interest	Yes/no for reporting of funding sources (and role of funders) and conflict of interest by authors
Protocol registration and publication	Yes/no for protocol registration and/or publication in peer-reviewed journal prior to the conduct of the review

**Table 2 ijerph-17-08658-t002:** Key data extraction elements for Systematic Review 2 (systematic review and meta-analysis in LMICs).

Data Element	Key Indicators
Publication data	First, journal, publication date
Study participants	Health data source/study population description, study/sampling period, geography (country or multi-country, region, state). maternal/neonatal factors (e.g., race/ethnicity, socioeconomic, marital status, comorbidities, sex and parity of birth), number of mothers and births (target, enrolled, follow-up rate, exclusion/inclusion criteria)
Methods	Study design, birth outcome (definition, assessment and prevalence or incidence in study population), exposure (sources and assessment methods, e.g., monitor, modelled, satellite imagery and hybrid method; and timeframe; e.g., whole pregnancy and trimester) and statistical methods
Results	Number of cases and controls in each study, exposure levels for each criteria air pollutant of interest and temperature (e.g., mean, median, quartiles/percentiles, range), main statistical findings (crude and adjusted effect estimates and reference unit with 95% confidence intervals and *p*-values for entire pregnancy period and by trimester or short-term and sex and any timeframe reported for the extreme temperatures), and adjustment of confounding factors (e.g., season of birth, pregnancy complications, smoking/alcohol, sociodemographic factors, infant’s sex, co-pollutant)
Conclusion, recommendations and limitations	Researchers’ summary statement/conclusion or interpretation of the main findings (particularly from the abstract), overall recommendations and limitations.
Funding and Conflict of interest	Yes/no for declaration of funding sources (and role of funders) and conflict of interest by authors

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
