# Peer review of "Ambient Air Pollution, Extreme Temperatures and Birth Outcomes: A Protocol for an Umbrella Review, Systematic Review and Meta-Analysis"

_ijerph, 2020, doi:10.3390/ijerph17228658_

Round 1

Reviewer 1 Report

Umbrella reviews represent one of the highest levels of evidence synthesis. The aim of an umbrella review is not to repeat the searches, assessment of study eligibility, assessment of risk of bias or meta-analyses from the included reviews, but rather to provide an overall picture of findings for particular questions or phenomenon.

It would be better if abstract will be divided into Background, Methods, and Implication sections.

The introduction provides a good, generalized background of the topic. Authors have included explanation of the topic and then provide context, and explain what are being challenged or extended to make the introduction more substantial.

Review objective(s) are clearly stated and logically inform the specific review questions.

Overall, this is a well written protocol. Authors may wish to use “This umbrella review will include…..” in the methods section to make it clearer for the readers.

Authors may wish to add ‘data extraction tool for systematic reviews and research syntheses’

Author Response

  1. “It would be better if abstract will be divided into Background, Methods, and Implication sections.”

Re: The abstract is presented in the journal’s style which requires structuring the abstract as Background, Methods, Results, and Conclusions but without headings.

  1. “Overall, this is a well written protocol. Authors may wish to use “This umbrella review will include…..” in the methods section to make it clearer for the readers.”

Re: The following statement has been added at lines 167-170 in page 4, blue track-changed: ‘This review will include a comprehensive synthesis of evidence from existing systematic reviews and meta-analyses through an umbrella review approach (systematic review 1) and systematically evaluate the primary evidence from LMICs (systematic review 2)’.  

  1. Authors may wish to add ‘data extraction tool for systematic reviews and research syntheses’

Re: Data extraction tools for both systematic reviews 1 and 2 are now added to the supplementary material in Table S3 and Table S4 and indicated in the main text in lines 322-324 (page 7); “Two authors will extract data from the selected studies with a data extraction tool (Table S3) developed according to the relevant data for the umbrella review as summarized in Table 1. The data extraction tool will be piloted prior to the full data extraction process” and 330-331 (page 8); “Similarly, data will be extracted by two authors with a piloted data extraction tool (Table S4) according to the key data elements for the systematic review 2 as summarised in Table 2.”

Reviewer 2 Report

Overall, this is a very well written manuscript. Based on the background and the provided study rationale, it will provides important results that are currently lacking in the existing scientific literature.

I have no revisions to indicate, the work is very clear and well explained.

Author Response

Has no comments to address.

Reviewer 3 Report

Overall this manuscript and methods are sound. I have a few general questions about methodology of the meta-analysis.

How will the authors account for publication bias? Journals tend to accept results with significant results while null results are less likely to be submitted or accepted for publication.

Including low and middle income countries in a meta analysis on this topic is of importance as these nations tend to have high air pollution than developed Western nations. These nations also tend to suffer from political instability, gender inequality, and nutritional deficits that are likely to impact birth outcomes. In the methods, the authors mentioned stratifying the studies by region and year. Is it possible to also include a measure of political stability or stratify studies by years from a major political event? The UN also publishes a gender inequality index. Could you stratify by gender inequality to examine potential confounding and/or effect modification by inequality?

Have the authors thought about methods to examine results by nutritional deficiencies or food shortages in the Low/Middle income countries studied? If so, please provide more details about the plan. If the authors do not think this is an issue of concern, please justify.

Similarly, how will the authors account for the wide variation in global climate and weather patterns?

Author Response

  1. “How will the authors account for publication bias? Journals tend to accept results with significant results while null results are less likely to be submitted or accepted for publication.”

Re: This has been outlined in the methods section at lines 426-431 under the subheading ‘Publication bias’ for systematic review 2. But this approach is not applicable/recommended for the umbrella review (systematic review 1) since we are not performing a meta-analysis of meta-analyses or statistically re-synthesising the results of the included primary studies. Publication bias is less observable among meta-analyses than primary studies. The background section of our manuscript supports this assumption, given we cited numerous published reviews that reported statistically non-significant findings.

2.”Including low and middle income countries in a meta-analysis on this topic is of importance as these nations tend to have high air pollution than developed Western nations. These nations also tend to suffer from political instability, gender inequality, and nutritional deficits that are likely to impact birth outcomes. In the methods, the authors mentioned stratifying the studies by region and year. Is it possible to also include a measure of political stability or stratify studies by years from a major political event? The UN also publishes a gender inequality index. Could you stratify by gender inequality to examine potential confounding and/or effect modification by inequality? Have the authors thought about methods to examine results by nutritional deficiencies or food shortages in the Low/Middle income countries studied? If so, please provide more details about the plan. If the authors do not think this is an issue of concern, please justify. Similarly, how will the authors account for the wide variation in global climate and weather patterns?”

Re: As suggested, we will stratify by i. gender inequality, ii) nutritional deficit status iii) political instability and iv) and climatic region; and this has now been specified in lines 413-419, page 10 which reads as follows:

“We will also perform subgroup analysis by World Bank’s economic group (low, lower-middle, and upper-middle)[70], global gender gap index on the scale of 0 (inequality) to 1(full gender equality) (0.0-0.2, worst; 0.3-0.5, bad; 0.6-0.8, good; 0.9-1.00, best) [89], country’s hunger and nutritional status with the global hunger index by severity scale (≤ 9.9, low; 10.0–19.9, moderate; 20.0–34.9, serious; 35.0–49.9, alarming; ≥50.0, extremely alarming) [90], political stability index [91] (≤ -2.0, extremely weak; -2.0 < to ≤ -1.0, very weak; -1.0< to ≤ 0.0, weak; 0.0 < to ≤1.0, moderate; 1.0 < to ≤2.0, strong; > 2, very strong), climatic zone (tropical, subtropical, temperate, polar/cold) [92].” We have also updated Table S4 accordingly.

Reviewer 4 Report

This is not a research article, and I do not find it interesting to read. Unlike a cohort study profile, which provides useful information to read, most of what I see in this article could be part of the supplemental information of future review articles that the authors are promising to conduct. I do not find this article interesting to read as a publication that does not have any result.

Author Response

“This is not a research article, and I do not find it interesting to read. Unlike a cohort study profile, which provides useful information to read, most of what I see in this article could be part of the supplemental information of future review articles that the authors are promising to conduct. I do not find this article interesting to read as a publication that does not have any result.”

Re: A number of methodological and reporting guidelines for reviews [1-4] recommended registration and/or publication of review protocols prior to the conduct of the review. Therefore, publication of this protocol ensures that we conform to best practice. Recent empirical evaluative studies [5,6] have demonstrated the usefulness of this practice in contributing to the transparency and credibility of review studies and subsequent findings. Published systematic review protocols, examples [7-9] are also relevant materials in the research community as valuable references for other review protocols and primary studies.

References

  1. Moher, D.; Shamseer, L.; Clarke, M.; Ghersi, D.; Liberati, A.; Petticrew, M.; et al. Preferred reporting items for systematic review and meta-analysis protocols (PRISMA-P) 2015 statement. BMC Rev. 2015, 4, 1,1
  2. Shamseer, L.; Moher, D.; Clarke, M.; Ghersi, D.; Liberati, A.; Petticrew, M.; et al. Preferred reporting items for systematic review and meta-analysis protocols (PRISMA-P) 2015: elaboration and explanation. BMJ. 2015, 350, g7647.
  3. Shea, B.J.; Reeves, B.C.; Wells, G.; Thuku, M.; Hamel, C.; Moran, J.; et al. AMSTAR 2: a critical appraisal tool for systematic reviews that include randomised or non-randomised studies of healthcare interventions, or both. BMJ. 2017, 358, j4008. The AMSTAR2 tool available at https://amstar.ca/docs/AMSTAR-2.pdf
  4. Joanna Briggs Institute. Joanna Briggs Institute Reviewer’s Manual. The Joanna Briggs Institute, Australia. Available at https://reviewersmanual.joannabriggs.org/.
  5. Ge, L.; Tian, J.; Li, Y.; Pan, J.; et al., Association between prospective registration and overall reporting and methodological quality of systematic reviews: a meta-epidemiological study J Clin Epidemiol. 2018,93:45-55. Doi: 10.1016/j.jclinepi.2017.10.012
  6. Sheehan, M.C.; Lam, J. Use of Systematic Review and Meta-Analysis in Environmental Health Epidemiology: a Systematic Review and Comparison with Guidelines. Curr Envir Health Rpt. 2015. 2,272–283. DOI 10.1007/s40572-015-0062-z
  7. Lopes-Júnior LC, Lima RAG, Olson K, et al. Systematic review protocol examining the effectiveness of hospital clowns for symptom cluster management in paediatrics. BMJ Open 2019;9:e026524. doi:10.1
  8. Lopes-Júnior, L.C.; Bomfim, E.; Silveira, D.S.C.da.; et al. Effectiveness of mass testing for control of COVID-19: a systematic review protocol. BMJ Open 2020;10:e040413. doi:10.1136/ bmjopen-2020-040413
  9. Silva Junior, F.J.G.da.; Sales, J.Ce.S.; Monteiro, C.Fd.S.; et al. Impact of COVID-19 pandemic on mental health of young people and adults: a systematic review protocol of observational studies. BMJ Open 2020;10:e039426. doi:10.1136/bmjopen-2020-039426.

Round 2

Reviewer 4 Report

If the authors search all databases of WOS, then MEDLINE and several others would be included. In general, I am still not a fan of such papers, though others have published similar study protocols. I believe this well fits mostly to the SI of future planned papers. No results to see so far, no guarantee for publication of future planned studies, so good luck with conducting your research!